# Ultrasonic Liquid Penetration Measurement in Thin Sheets—Physical Mechanisms and Interpretation

**DOI:** 10.3390/ma13122754

**Published:** 2020-06-17

**Authors:** Carina Waldner, Ulrich Hirn

**Affiliations:** 1Institute of Bioproducts and Paper Technology, TU Graz, Inffeldgasse 23, 8010 Graz, Austria; carina.waldner@tugraz.at; 2CD Laboratory for Fiber Swelling and Paper Performance, Inffeldgasse 23, 8010 Graz, Austria

**Keywords:** liquid penetration, ultrasound transmission, capillary penetration, porous sheets

## Abstract

Ultrasonic liquid penetration (ULP) measurements of porous sheets have been applied for a variety of purposes ranging from determining liquid absorption dynamics to surface characterization of substrates. Interpretation of ULP results, however, is complex as the ultrasound signal can be affected by several mechanisms: (1) air being replaced by the liquid in the substrate pores, (2) air bubbles forming during penetration, and (3) structural changes of the substrate due to swelling of the substrate material. Analyzing tailored liquids and substrates in combination with contact angle measurements we are demonstrating that the characteristic shape of the ULP measurement curves can be interpreted in terms of the regime of liquid uptake. A fast and direct decline of the curve corresponds to capillary penetration, the slope of the curve indicates the penetration speed. A slow decline after a previous maximum in the signal can be related to diffusive liquid transport and swelling of the substrate material.

## 1. Introduction

Ultrasound attenuation measurements during liquid penetration have originally been reported to be able to determine the degree of sizing in paper [1] and correlations to liquid absorption measurements (Bristow absorption test) have been found [2]. Since then, ultrasonic liquid penetration (ULP) measurements have been applied to study different aspects of interactions between liquids and porous, sheet like materials (paper, membranes, and thin films). Things investigated were for example network hydrophobization (sizing) [3,4,5,6], kinetics of liquid absorption (wetting and penetration) [7,8,9,10], evaluating water/grease barrier properties [11,12,13,14], as well as predicting print quality [15,16,17,18] and gluing strength [19]. The method has even been used to characterize the structure of the paper substrate, e.g., surface pores and roughness [20,21,22,23].

In these publications, a wide range of liquids and substrates were combined, ranging from all kinds of papers to membranes and films on the substrate side and from water, to alcohols, oils, glues, coating colors, inks, and all kinds of mixtures on the liquid side. There is hardly any limit to combinations of substrates and liquids that can be combined and investigated with ULP. The only requirement for a sample to be measurable seems to be that attenuation is not too high so that a signal can still be detected at the receiver. This is affected by a combination of factors like sample thickness, porosity, pore size, and other material properties like density and elasticity. Furthermore, the signal frequency is playing a large role. For samples measured with 2 MHz, thickness was between 40 µm and 2 mm, pore size was between 0.01 and 100 µm, and porosity was between 2% and 90%.

The measurement principle is shown in Figure 1a. The sample is fixed on a sample holder with two-sided adhesive tape. When the test is started, the sample holder is quickly immersed into the test liquid. As soon as the sample is completely immersed, a high frequency (typically 1 or 2 MHz), low energy ultrasound signal is transmitted through the sample in thickness direction and the changes in transmission are recorded over time. The signal is a pulse with an initial pulse length in the sender of about 1 µs. At the receiver, the pulse builds up over seven oscillations and declines over seven oscillations. Consequently, no difference compared to a continuous wave regarding the interference should be seen in practice [24].

Figure 1c shows two types of sample holders. On the flat sample holder (left) the sample backside is completely covered with the tape, while the sample is not in contact with the tape in the grooves of the grooved sample holder. Unless otherwise stated measurements discussed in this paper were made with the flat sample holder.

Figure 1b shows two typical ultrasound transmission curves of water and an isopropanol–water mixture on a sized copy paper on the flat sample holder. The curves show the ultrasound transmission intensity over time, as a percentage of the maximum ultrasound signal collected. Thus, only the changes of the signal during penetration affect the shape of the curve. In Figure 1b, the curve for water initially exhibits an increase in intensity, followed by a decline. The curve for isopropanol/water immediately declines rapidly. In order to be able to interpret these curves, it is necessary to understand mechanisms that influence the signal during liquid penetration. Several mechanisms are said to have an impact on the signal during liquid penetration (compare for instance [5,6,15,22]):Air being replaced by the liquid;Air bubbles forming during penetration;Structural changes of the substrate (paper) due to swelling of the fibers.

However, it is not very well understood how these processes affect the signal transmission, how the individual mechanisms are related to the shape of the measurement curves, and which of the mechanisms has the strongest impact. Without this understanding, interpretation of results is difficult. For instance, the time to the maximum intensity has been interpreted as the wetting time (e.g., [3,5,15]), but no correlation with wetting measured via contact angle measurements could be found [8]. Therefore, the aim of this paper is to clarify how these mechanisms influence ultrasound transmission, which of the mechanisms is dominant in certain circumstances and finally, to provide a guideline for interpretation of the result curves. To be able to improve the understanding of how the individual mechanisms influence ultrasound transmission during liquid penetration, first, fundamentals of ultrasound transmission and liquid penetration are discussed. Then we will discuss the individual physical mechanisms leading to different shapes of the ultrasound intensity curves, and provide measurements confirming that the measured ultrasound intensity is indeed linked to the liquid penetration regime into the structure. Finally, we describe different parameters describing liquid penetration in thin substrates obtained from ULP measurements.

## 2. Fundamentals of Ultrasound Transmission

On its way through the sample, the ultrasound wave is attenuated, i.e., the amplitude and ultrasound intensity decrease. The ultrasound intensity is defined as the energy flow rate transmitted through a unit area perpendicular to the propagation direction and is proportional to the square of the amplitude [25]. Attenuation is mainly caused by thermal and viscous losses, reflection occurring at phase boundaries, and scattering. Thermal and viscous losses have only a negligible impact in the measurement setup shown in Figure 1a [24]. Since the signal attenuation within a material is not relevant, attenuation is dominated by reflection and scattering at material interfaces.

### 2.1. Attenuation Due to Reflection

Attenuation of the ultrasound signal due to reflection at phase boundaries occurs if two adjacent media have differing wave impedances. The larger the wave impedance difference, the more reflection occurs. The wave impedance can be calculated for many materials with: [26]
Z = ρ × c,(1)

With Z being the wave impedance (kg/m²s), ρ the material’s density (kg/m³), and c the speed of sound in the material (m/s). For the case of perpendicular incidence of an ultrasonic compressional wave the reflection coefficient R at the boundary is: [26]
(2)$$R=\frac{Z_{1}{\;-\;Z}_{2}}{Z_{1}{+Z}_{2}}\;$$

Z_1_ and Z_2_ being the wave impedances of the two media. Equation (2) shows that less reflection occurs if the wave impedances of the two media are similar. Table 1 shows the wave impedances for air, water, and pulp fibers in paper calculated from density and speed of sound. During the penetration of a paper sample with water, the ultrasound signal can be reflected at the boundaries between air and water, air and fibers, and water and fibers. From the wave impedances, the reflection coefficient at each type of boundary can be calculated (see Table 2).

The reflection coefficients show that at an air-fiber boundary most of the signal was reflected. The same is true for air–water boundaries. If the air within a paper sample is completely replaced by water, water–fiber boundaries are replacing the air-fiber boundaries. The reflection coefficient for water-fiber boundaries was much lower, yielding a better signal transmission. This means that in general the received ultrasound signal should increase when the air in the sample is replaced by the test liquid during penetration. Please note that the wave impedance value for fibers should not be understood to be valid for all paper fibers, since the speed of sound in the fibers largely depends on the manufacturing process (compare [29]). 

The system of air, water, and pulp fibers is used as an example for a liquid penetrating into a porous material. Since solids and liquids generally have more similar wave impedances compared to gases [27], the considerations made above are also valid for other porous solids being penetrated by any kind of liquid. In conclusion, displacement of air by liquid in the substrate pores thus can only be responsible for an increase in ultrasound intensity over time, the curves however often show a (pronounced) decrease.

### 2.2. Attenuation Due to Scattering

Liquid penetrating into a porous substrate represents a very inhomogeneous system in which water, fibers/pores, and air bubbles are distributed irregularly. In such an inhomogeneous system, scattering of the ultrasound signal influences the received signal [27]. Since the transmitted ultrasound intensity is normalized in the results, only factors that could change the scattering behavior during penetration are relevant.

As argued above, the air being replaced by the liquid should generally lead to a signal increase due to the lower reflection coefficient of a solid–liquid boundary. However, if the air cannot escape during penetration, air bubbles could be entrapped within the porous substrate and constitute additional scattering centers, ultimately leading to a signal decrease. Daun [24] showed how bubbles of different sizes impact scattering (see Figure 2). The scattering cross section thereby refers to the ratio of the total scattered power and the intensity of the incoming signal. The total scattered power is obtained by integrating the scattered intensity in each direction over the surface of a sphere. Taking resonance of bubbles into account, small bubbles of a critical size can increase scattering effects by several orders of magnitude. Figure 2 also shows that resonance effects depend on the frequency of the ultrasound signal. For 1 MHz the scattering peak is at a bubble radius of about 3.2 µm, for 2 MHz at 1.5 µm. Daun concluded that “the scattering due to resonance is the dominant mechanism for scattering”.

For ULP measurements this means that the signal will be affected strongly if air bubbles of a critical size are formed and entrapped during penetration. Those entrapped bubbles scatter the signal and lead to a sharp decrease in signal transmission. First, the pore size of the substrate limits the maximum possible bubble size and thus also determines if bubbles of a critical size will be created. Second, the degree of scattering always also depends on the measurement frequency as the resonance shows sharp peaks for certain bubble size–frequency combinations.

## 3. Fundamentals of Liquid Penetration 

In order to be able to understand mechanisms that influence the ultrasound signal during penetration, a fundamental understanding of liquid penetration dynamics is necessary. In general, penetration of a liquid into a porous substrate takes place by capillary flow into the pores. Assuming cylindrical pores, the flow can be described with the Lucas–Washburn equation. It can be derived from the Poiseuille equation for laminar flow: [30]
(3)$$\frac{\mathrm{dh}}{\mathrm{dt}}=\frac{r^{2}\Delta p}{8\eta h}\;,$$
where h (m) is the distance travelled by the liquid, r (m) the capillary radius, η (Ns/m²) the liquid viscosity, and ∆p (N/m²) the driving pressure difference. If gravity effects are negligible—as is the case for small pores in the micrometer range like in paper [31]—the driving pressure difference is the capillary pressure. It can be described with the Laplace equation: [32]
(4)$$\Delta p=\frac{{2\gamma }_{L}\cos\theta }{r}\;,$$

With γ_L_ (N/m) being the surface tension of the liquid and θ (°) the contact angle of the liquid with the solid. Substituting Equation (4) into Equation (3) and integrating over time leads to the Lucas–Washburn Equation:(5)$$h=\sqrt{t}\;\cdot \sqrt{\frac{{r\gamma }_{L}\cos\theta }{2\eta }}$$

This equation indicates two important relations. First, it shows that the penetration depth (and thus the penetration volume) is proportional to the square root of time. Second, the penetration is driven by the contact angle between the liquid and the solid. Equation (4) shows that the capillary pressure is only positive if cos θ > 0, i.e., θ < 90°. If θ > 90° the liquid is not sucked into the capillaries, but instead an external pressure needs to be applied so that the liquid enters the pores. Therefore, the contact angle between the liquid and the solid is the key criterion to determine if capillary penetration is possible.

Penetration into a real porous substrate is much more complex than penetration into cylindrical pores. Nevertheless, the Lucas–Washburn equation well illustrates the fundamental requirements for penetration in porous materials.

If capillary penetration is not possible, the liquid in contact with a porous substrate can still enter the pores as vapor, via surface wetting, by penetration within the porous paper fibers, or by diffusion. That is especially relevant if the substrate interacts with the liquid, as is the case for aqueous liquids and paper. The transport of water vapor through paper is most likely primarily driven by surface diffusion [30]. That means that water can enter the paper also if it is heavily sized and the contact angle is more than 90°. However, in that case there will be no liquid front filling the pores as is the case for capillary penetration. Instead water molecules will diffuse along or within the fiber surface, causing them to swell.

## 4. Changes in Ultrasound Transmission During Liquid Penetration

Based on the theoretical considerations made above, three cases of liquid penetration can be distinguished, each yielding a characteristic ULP curve shape. First, we distinguished between signal increase and decrease. As indicated earlier, a signal increase would be the general case to be expected if the air in the sample is displaced by the test liquid. A decrease can occur in certain cases due to entrapped air bubbles or swelling of the fibers.

### 4.1. Case 1—Signal Increase

A signal increase can be explained by a decrease of the reflection coefficient if the air in the sample is homogeneously replaced by the test liquid. The ultrasound signal is reflected to a large extent at air-solid boundaries and air-liquid boundaries, but much less at liquid-solid boundaries. When the sample is being filled with liquid, less and less of the signal is reflected and thus, the signal increases. 

Figure 3 illustrates a liquid penetration process yielding an increasing ultrasound signal. The exemplary ULP curve is measured with a hydrophilic polyethylene membrane in contact with water. At the start the ultrasound signal is reflected to a large extent due to an air film present at the surface of the substrate (Figure 3b—image 1). When the sample is wetted by the liquid, the air film at the surface disappears and thus less reflection occurs (Figure 3b—image 2). Once the sample is soaked by the liquid, less and less reflection occurs, and the signal continues to increase. Since the sample backside is closed, the air within the sample possibly cannot escape as it is pushed back by the penetrating liquid front. Air bubbles may form. However, if the bubbles are not in a size range critical for resonance, they will not stop the signal increase. For non-resonating air bubbles to lead to a signal decrease, more air-liquid boundaries would have to form than air-solid boundaries were present at the beginning. That is not to be expected.

As mentioned earlier, cases with and without capillary penetration should be distinguished. A signal increase can occur in both cases. The starting situation for both cases is the same. However, if capillary penetration is not possible, the liquid will not penetrate the substrate and thus not advance any further than in Figure 3b—image 2. Since the air film at the surface of the sample is removed nonetheless, the signal will still increase at the start also without capillary penetration. An example of such a case is a CaCO_3_-polyethylene composite in contact with water (see Section 5). 

### 4.2. Case 2—Signal Decrease Due to Critical Air Bubbles

In general, one would expect the ultrasound signal to increase during penetration. However, some substrates exhibited a steep decrease of the ultrasound signal immediately after getting into contact with the liquid. This is often the case for substrates where capillary penetration is possible and takes place rapidly. Such a behavior can be explained by air bubbles of a critical size. As mentioned before, if the air within the sample cannot (completely) escape, air bubbles will form during penetration. If the size of the air bubbles is in a critical range for resonance, they will drastically reduce ultrasound transmission (compare Figure 2) and cause the sharp decrease in transmission.

This behavior is typical for hydrophilic (unsized) paper in contact with aqueous liquids. Figure 4 illustrates what happens during capillary penetration of such a substrate. Wetting of the sample was so fast that it could not be detected by the ULP measurements. Penetration took place rapidly and air bubbles resonating with the signal form immediately, leading to a steep decrease of the ultrasound signal.

In order to prove that the sharp decrease of transmitted signal is indeed caused by enclosed air bubbles, we compared measurements of hydrophilic paper with water, performed with two different sample holders. The ULP measurements discussed so far were all performed with a flat sample holder on which the samples were attached via two-sided adhesive tape (compare Figure 1c left). In this case the air cannot escape at the backside of the sample and was trapped within the substrate, where air bubbles started to form. A sample holder with grooves allowed the air to escape at the backside of the sample (compare Figure 1c right). The adhesive tape was pushed into the grooves with a grooved cylinder. Thus, the sample was in contact with the tape only at the peaks between the grooves. Still, the sample was in contact with the liquid only on the top side. When the liquid then penetrated the sample, the air was again pushed to the back, but could now escape via the grooves.

Figure 5 compares ULP measurements of unsized paper on the flat and grooved sample holder. On the flat sample holder (Figure 5a) the air could not leave the sample and air bubbles were entrapped while the air could leave on the grooved sample holder (Figure 5b). The shape of the two curves was remarkably different. While the signal decreased strongly due to the air bubbles formed in the case of the flat sample holder, there was a steep increase of transmission when the air was allowed to escape. This means that without entrapped resonating air bubbles, the signal increased due to the air being replaced by the liquid, as predicted in case 1. For the case of capillary penetration together with resonating air bubbles on a flat sample holder, on the other hand, the changes in signal transmission were controlled by the rate of formation of entrapped bubbles.

### 4.3. Case 3—Signal Decrease Due to Swelling

If capillary penetration is possible, a signal decrease is caused by entrapped, resonating air bubbles, as explained above. However, paper samples where capillary penetration is not possible because they are heavily sized (θ > 90°) can still exhibit a decrease in signal transmission. Air bubbles cannot be the reason for the decrease in this case, as there is no liquid entering the substrate pores. This can be shown when again comparing the flat and the grooved sample holder. Figure 6a shows the ULP curve of a sized copy paper with water on the flat sample holder, where the air within the sample is entrapped at the backside. The signal first increased until it reached a maximum that was followed by a slow decrease. The shape of the curve was not changed when using a grooved sample holder that allows the air to escape (Figure 6b). Thus, enclosed air bubbles could not be the reason for the signal decrease in this case.

Air bubbles cannot be the reason for a signal decrease if capillary penetration is not possible, since there is no liquid front advancing through the sample. However, as mentioned earlier, water can still enter the sample via vapor and surface diffusion, as well as penetration within the paper fibers (which are porous). In that way, water will diffuse into the fibers, filling voids within the fibers, causing them to swell. The filling of the voids should lead to an increase in the speed of sound in the fibers. That would lead to an increase in the fibers’ wave impedance and consequently to an increased reflection coefficient for air–fiber boundaries. As the fibers are still surrounded by air, this would explain the slowly decreasing ultrasound signal.

Figure 7 shows the last stage of penetration of water into a sized paper. The first two stages were equivalent to the first two stages of case 1 (Figure 3—top and middle row). Initially there was an air film at the paper surface at which the ultrasound signal was reflected to a large extent. As the liquid wetted the surface and the air film disappeared, ultrasound transmission increased. For a non-swelling substrate–liquid combination the signal would not decrease again. For paper in contact with a fiber swelling liquid like water, however, the fibers will take up some liquid, ultimately leading to a decrease in the signal. In that case, the time of the maximum intensity can be interpreted as the time at which swelling of the fibers starts [33].

Results that confirmed that fiber swelling was the reason for the decrease in ultrasound transmission for surface hydrophobized (sized) papers are shown in Figure 8. First, Gabriel [6] found that ultrasound transmission and wet expansion are strongly correlated especially for sized and coated papers (Figure 8a) and that they show exactly the same development over time. The wet expansion of a paper sheet is directly related to the swelling of the fibers within. Therefore, this correlation is a strong indication that the decrease of the ULP curve for hydrophobized papers is indeed related to swelling of the fibers.

Second, Gabriel [6] also found that the ULP results are highly dependent on the temperature of the test liquid. In Figure 8b 20 percent ultrasound intensity was reached after about 14 s when measuring water at 23 °C, while it took less than half the time to reach the same relative intensity at 43 °C. The faster decrease at higher temperatures indicates faster water uptake of the fibers. One reason for that might be the lower viscosity of the liquid. More dramatic, however, is the increase in saturation vapor pressure, which increased exponentially with temperature. Therefore, Gabriel [6] concluded that the sharper decrease and thus faster liquid uptake at higher temperatures is caused by the intensified water vapor diffusion into the sample and the fibers. This is another indication that the decrease of the ULP curve of a surface hydrophobized (sized) paper is determined mainly by the transport of the liquid into the fibers.

## 5. Impact of Penetration Mechanism on Curve Shape

As discussed above, ultrasound transmission is influenced in different ways depending on whether capillary penetration of the liquid in the substrate is possible or, if not, by swelling of the substrate (often paper fibers). Each of the three discussed cases leads to a characteristic curve shape. In order to prove that capillary penetration and fiber swelling are determining the shape of the ULP curves, measurements with different test liquids were performed on two substrates. The liquids were designed so that some of them would penetrate via capillary penetration and some not. One of the substrates contains fibers that swell when in contact with aqueous liquids, the other not. 

### 5.1. Materials and Methods

Table 3 gives an overview of the test liquids used. They are all based on deionized water with additives to modify the penetration behavior. Additives used were glycerol (AnalaR® NORMAPUR® from VWR, 99.5%), hexanediol (1,2-Hexandiol, Alfa Aesar from Thermo Fisher Scientific, 97%), and isopropanol (EMSURE® ACS from Merck, 99.8%). Glycerol was used to change the contact angle of the liquid, hexanediol and isopropanol also lowered the surface tension, which also affects the contact angle. Naphthol Blue Black (98%) was used as a dye for better visibility, but does otherwise not change the liquid properties (compare [34]). In that way, the liquids were tuned so that some of them penetrate via capillary penetration and others not. To determine whether capillary penetration is possible for a liquid–substrate combination, contact angle measurements were performed with a Fibro DAT 1100 dynamic contact angle instrument from FIBRO System, Netherlands, according to ANSI/Tappi standard T 558 om-15. Twenty individual drops with a volume of 4 µL were measured for each liquid–substrate combination. The contact angle was measured as soon as the drop had completely settled at the surface (20 ms or 40 ms after first drop-surface contact). The non-swelling substrate is a porous composite consisting of 80 percent calcium carbonate and 20 percent polyethylene. The swelling substrate is an AKD sized standard all-purpose office paper, consisting of bleached eucalyptus pulp and 14 percent calcium carbonate as filler. ULP measurements were performed with a Penetration Dynamics Analyser 2.0 from emtec Electronic GmbH, Leipzig, Germany, using a measurement frequency of 2 MHz and a flat sample holder. The samples were cut to rectangles of 7 cm × 5 cm and attached to the sample holder with two-sided adhesive tape. The ultrasonic sensor area used is a circle with a diameter of 35 mm. Each curve presented has been averaged from four individual measurement curves. Measurement time was set to 5 s.

### 5.2. Results and Discussion

The CaCO_3_-polyethylene composite will not swell when in contact with any of the test liquids. Therefore, in theory, only the first two cases discussed before can affect the ultrasound signal. That means that in general the signal should increase during liquid contact (case 1). A decrease is only possible if the sample is penetrated via capillary penetration and air bubbles of a critical size for resonance form (case 2).

Figure 9b shows that the test liquids 3 and 4 as well as deionized water had contact angles of more than 90° on the CaCO_3_ -polyethylene composite sheets. That means that capillary penetration was not taking place for those liquids. The ultrasound transmission curves (Figure 9a) for these three liquids had a similar shape and all increase continuously throughout the measurement (i.e., case 1). This is what would be expected of liquids without capillary penetration on a non-swelling substrate.

The center point test liquid (TLC) and the isopropanol–water mixture (IPA) had contact angles of less than 90° and thus, capillary penetration was possible. In contrast to the other liquids, ultrasound intensity decreased for those two liquids. This indicates that air bubbles formed during the penetration of the substrate, which resonate with the signal and attenuate it. Furthermore, the TLC liquid had a higher contact angle than the IPA liquid. This is in line with the ultrasound transmission decreasing more slowly for the TLC than the IPA liquid. For a higher contact angle, the capillary pressure was lower and thus also the driving force for penetration was reduced (compare Section 3), ultimately leading to slower attenuation of the ultrasound signal.

From the measurements on the CaCO_3_ polyethylene composite sheets it could be concluded that the shape of the ultrasound transmission curve is an indicator for the penetration mechanism taking place for non-swelling substrates, if bubbles of a critical size are formed during penetration. In the case of capillary penetration with critical bubble formation, the slope of the curve is an indicator for penetration speed.

As discussed before, the ultrasound signal can be affected also by changes in the fibers due to liquid uptake when it comes to paper in combination with fiber swelling liquids (case 3). This effect is especially relevant for sized papers, where the fiber surface is chemically hydrophobized, which hinders capillary penetration. Figure 10b shows that again test liquids 3 and 4, as well as water formed contact angles of more than 90° on the AKD sized paper. The TLC liquid was at the very edge of the 90° line. The shape of the ultrasound transmission curves (Figure 10a) of these four liquids matched the curve described as case 3. The increase of the ultrasound transmission at the beginning was followed by a decrease caused by diffusive liquid uptake of the fibers. For TL3 and TL4 the decrease was not very evident because it was rather slow. It would be more prominent at longer measurement intervals. 

TL3 exhibited the highest contact angle and was also the last to reach its maximum in the ultrasound transmission. The lower the contact angle of those four liquids, the better is the surface wetting of the liquid and the faster the maximum of the curve is reached. TLC, which was at the very border of the 90° limit, was the first to reach its maximum in ultrasound transmission and the signal decreased faster after the maximum than for the other three liquids with θ > 90°. This indicates a faster liquid uptake of the fibers.

The other three test liquids—TL1, TL2, and IPA—had contact angles below 90°. For these liquids capillary penetration took place and they exhibited a curve shape, which matched case 2—capillary penetration with critical bubble formation. TL2 and IPA had a much lower contact angle than TL1 and demonstrated a steeper decrease in ultrasound transmission as well. Just like for the liquids with capillary penetration on the CaCO_3_ polyethylene composite sheets, the slope of the curve is an indicator for penetration speed also for paper if capillary penetration takes place. 

These measurements suggest that the shape of the ULP curve can be an indicator for the dominating liquid uptake mechanism for the paper. However, swelling of the fibers could also have a (minor) impact if capillary penetration is possible. There might be a transitional area close to contact angles of 90° in which both resonating air bubbles and swelling fibers affect ultrasound transmission equally. Furthermore, if paper properties vary significantly throughout the thickness direction (like is the case for highly calendared or coated papers) interpretation of ULP signals becomes even more complex. In order to avoid incorrect interpretations of ULP measurements we recommend to always perform complementary contact angle measurements.

## 6. Parameters for Evaluation of ULP Measurements

After elucidating the interpretation of the ULP measurement curves, it is useful to define parameters that describe specific aspects of the liquid–substrate interaction. In literature a huge variety of parameters is used. Most of the parameters used refer to curve shapes of case 3—curves with a maximum after which the signal declines. Some parameters describe the first part of the curve, which are said to describe surface wetting and surface hydrophobization: L = intensity difference between the initial and the maximum signal [15,16];G1 = slope of the curve before the maximum [16];W = area above the curve until the maximum [4,23].

The parameter used most often for this type of curve is the time until the maximum intensity is reached (t_max_) [3,4,5,8,11,12,13,14,15,16,23,35], which often has been interpreted as the wetting time. As we pointed out, it is rather related to the onset of fiber swelling. To describe the curve after the maximum, several parameters related to the slope of the curve were used, which were interpreted as penetration speed and thus to be related to (bulk) hydrophobicity and porosity:A30/A60 = area above the curve after maximum until 30/60 s measurement time [3,4,23];Slope after maximum [8,11,12];T95/T60 = time until ultrasound intensity declined to 95/60 percent (after maximum) [5,13,21,22].

For the curves of case 3, we proposed to use the time until the maximum intensity to describe the first part of the curve as a measure for the time after which fibers begin to swell. The slope of the curve or a time until a certain intensity was reached (e.g., T95) should be used for the second part of the curve (Figure 11a). For a hydrophobic paper, the slope as well as T95/T60 describe the speed of water uptake of the fibers via vapor and surface diffusion.

When capillary penetration takes place and the resulting curves do not show a maximum (case 2), we recommend to use the same parameters to describe the slope of the curve as for the decreasing branch of curves with maximum (Figure 11b), i.e., T95/T60 or the slope of the curve. These parameters are a measure for the penetration speed in this case. When calculating the slope of the curve, it is important to choose a useful time interval for which the slope is calculated. Figure 11b illustrates a poorly chosen time interval. It is so large that in that interval the slopes of the two blue curves are equivalent although the dark blue curve is decreasing faster. Therefore, the time until a certain ultrasound intensity is reached (T95/T60) seems to be the more reliable parameter.

## 7. Recommendations

In addition to the parameters to be used, there are some general recommendations when performing ULP measurements. First, as mentioned before, we proposed to perform contact angle measurements with the same liquids and substrates in order to find out whether capillary penetration is taking place for a liquid substrate combination. Measuring the contact angle will support interpretation of the ULP results.

Second, we wanted to point out the necessity of understanding the processes taking place during penetration in order to enable a sound interpretation of the results. If new combinations of liquids and substrates are used, it might not be clear which physical/chemical processes could influence ultrasound transmission during penetration. There might be cases in which the slope of a decreasing ultrasound curve cannot be simply interpreted as penetration speed. An example for such a case would be measurements with coagulating inks on primed surfaces, where the ink forms a coagulating layer once it gets into contact with the paper surface.

If ULP measurements of new combinations of liquids and substrates are to be performed, a comparison of measurements with a flat and a grooved sample holder might be useful. Comparing the two sample holders can reveal how strongly ultrasound transmission is influenced by entrapped air bubbles, which is an important information for the interpretation of results.

Furthermore, mostly 1 and 2 MHz have been used to perform ULP measurements and these frequencies have proved to be suitable for paper. However, other measurement frequencies could potentially reveal events on different spatial scales [10] and might be needed for substrates other than paper (e.g., membranes, nanoporous films, etc.). In this context it is important to remember that the bubble size critical for resonance also depends on the frequency of the ultrasound signal. 

## 8. Conclusions

The main mechanisms influencing ultrasound transmission during liquid penetration were examined in order to deliver a basis for sound interpretation of ULP measurement curves. Three cases could be distinguished each yielding a characteristic curve shape:In general, air being replaced by a liquid in the sample would lead to a continuous increase in ultrasound signal due to a reduction of ultrasound reflection at air–substrate boundaries.If capillary penetration of the liquid into the substrate was taking place (indicated by a contact angle lower than 90°), a sharp decrease of ultrasound transmission was often observed. This was caused by enclosed air bubbles forming during penetration. If their size was in a range critical for resonance effects, the ultrasound signal was heavily scattered and thus attenuated. Therefore, the ultrasound transmission curve in this case reflected the rate of air bubble formation during penetration, which was directly related to the penetration speed of the liquid into the paper.If capillary penetration could not take place, air bubbles could not form, and the signal normally increased continuously as the air layer between liquid and substrate disappeared. For swelling substrates like paper, however, another mechanism was taking place. The typical measurement curve exhibited an increase up to a maximum followed by a decrease in ultrasound transmission. The signal first increased due to liquid replacing the air layer at the paper surface. After some time, the substrate (fibers) started to take up water via vapor and surface diffusion, which caused a density increase and thus higher reflection at the air–fiber interfaces, which resulted in a decreasing signal. The decrease in ultrasound transmission was proportional to water uptake of the fibers in this case. This means that the time until the maximum was related to the onset of fiber swelling. A shorter time to maximum and faster decrease after the maximum means faster swelling.

With testing liquids tailored to show (or not show) capillary penetration, as indicated by the contact angle, we were able to prove that a fast falling ULP curve without a maximum indeed indicates the presence of capillary penetration in the substrate. By comparing the curves between non-swelling and swelling substrates in the absence of capillary penetration we could demonstrate that the water uptake in the substrate material (in this case the liquid uptake into the paper fibers due to diffusion and intra-fiber wetting) corresponds to the slope of the signal decrease after the maximum in the curve. These results suggest that ULP measurements together with contact angle measurements could be a means to quantify first capillary penetration into substrate pores, and second diffusion induced liquid uptake in the substrate material (in the absence of capillary penetration).

## Figures and Tables

**Figure 1 materials-13-02754-f001:**
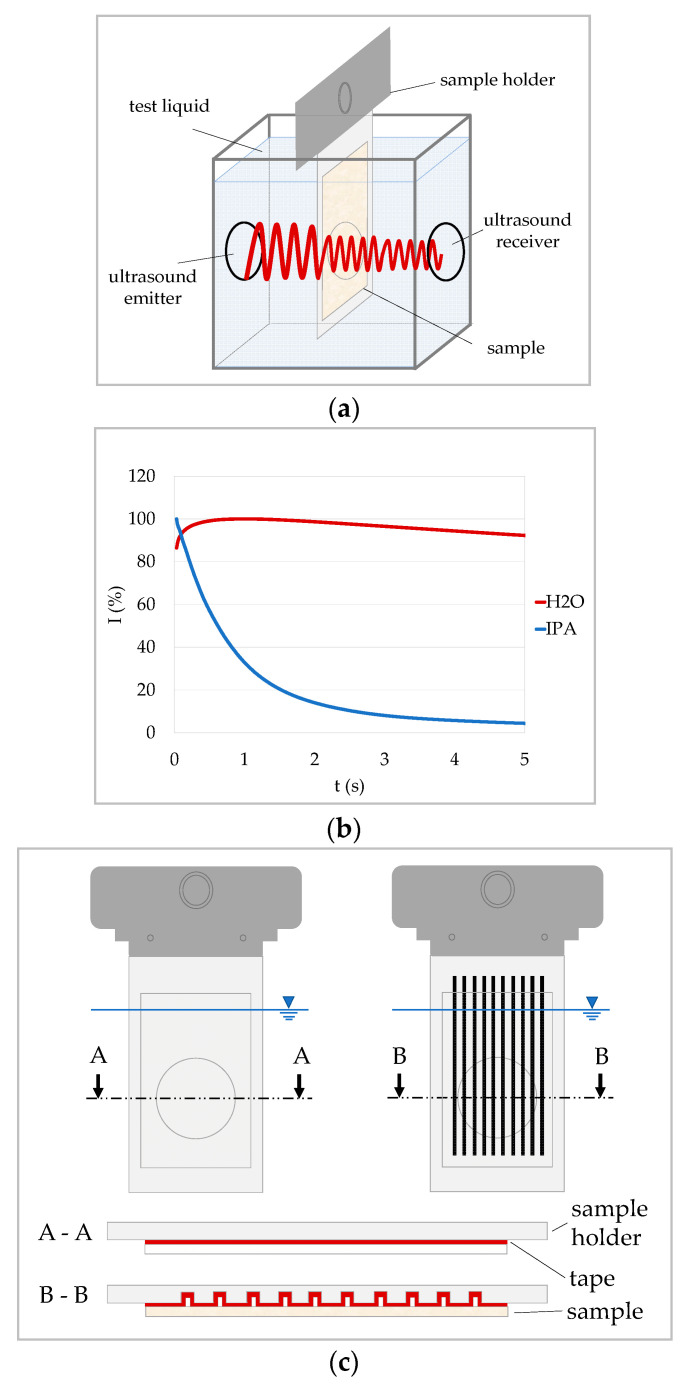
Ultrasonic liquid penetration (ULP) measurement: (**a**) measurement principle: the sample is immersed in the liquid; ultrasound is transmitted through the sample in thickness direction and the transmitted signal intensity I is recorded over time; (**b**) typical ULP curves of water and an isopropanol-water mixture (IPA) on sized copy paper on flat sample holder showing the relative change of the ultrasound intensity at the receiver; the two liquids lead to different curve shapes; and (**c**) different types of sample holders: smooth sample holder (left), sample holder with grooves (right; drawing not to scale).

**Figure 2 materials-13-02754-f002:**
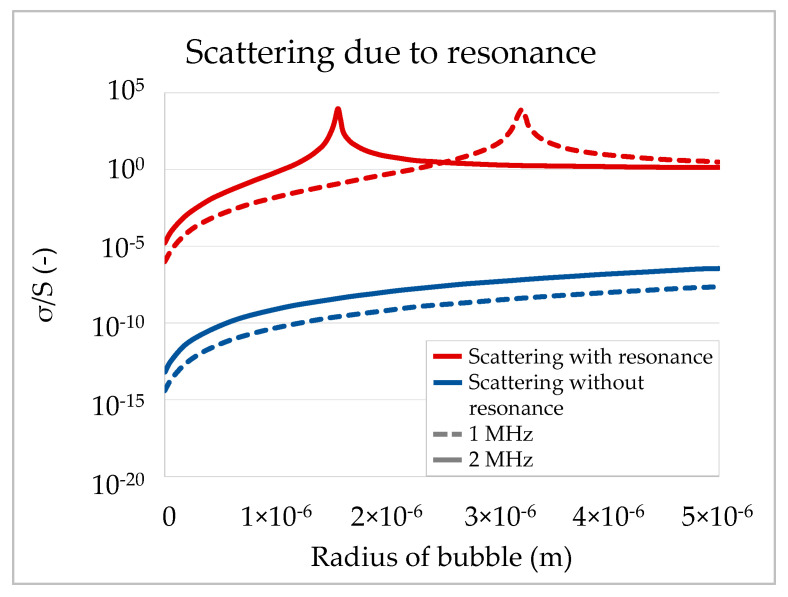
Scattering cross-section σ in relation to the bubble’s geometric surface S as a function of the bubble radius. Resonance increases scattering effects by several magnitudes for bubbles of a critical size (adapted from [24]).

**Figure 3 materials-13-02754-f003:**
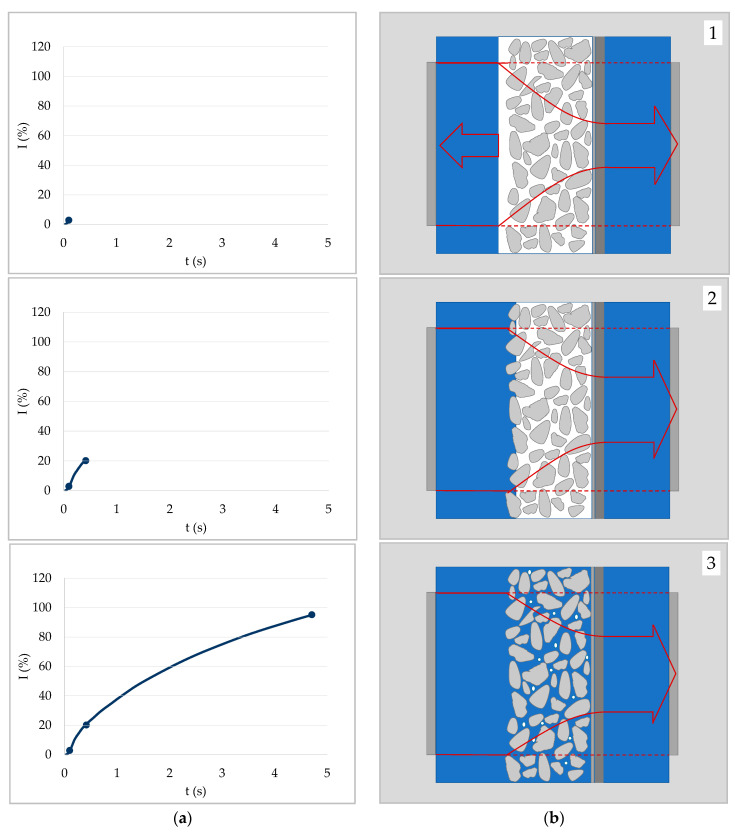
General case of a liquid penetrating into a porous substrate. No swelling or critical air bubbles occur. (**a**) The ULP curve of a hydrophilic polyethylene membrane in contact with water. (**b**) Graphical illustration of three stages during liquid penetration of the substrate as a function of time. Air film between liquid and surface (1), surface wetting of substrate (2), and full capillary penetration of the substrate (3).

**Figure 4 materials-13-02754-f004:**
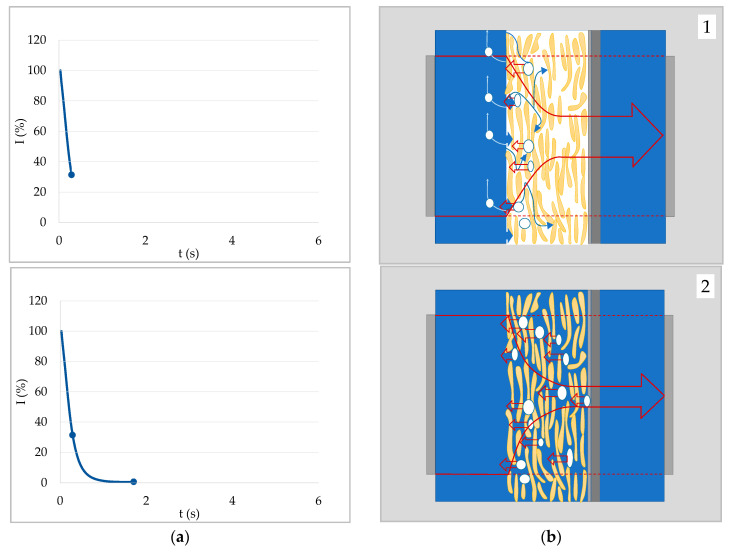
Capillary penetration of a liquid into a substrate (paper) with critical air bubbles being formed. (**a**) ULP curve of an unsized paper in contact with water. (**b**) Graphical illustration of two points in time during liquid penetration of the unsized paper: immediate wetting of the surface (1), air bubbles trapped in the substrate lead to ultrasound scattering, and a fast decrease of the signal (2).

**Figure 5 materials-13-02754-f005:**
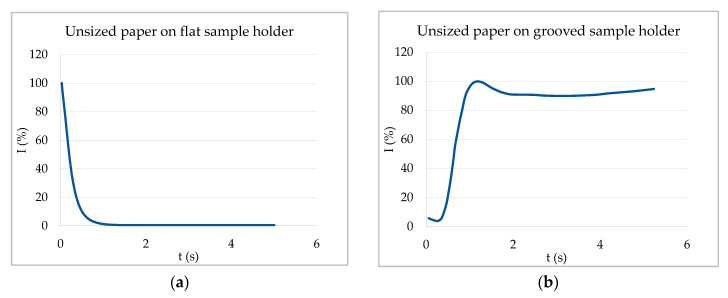
ULP curves of unsized paper in contact with water. (**a**) Unsized paper on a flat sample holder—air bubbles are entrapped leading to a sharp decrease of the signal. (**b**) Unsized paper on grooved sample holder (measurement from [16])—air can leave the sample on the backside via the grooves. Thus, no air bubbles form and the signal increases due to filling of the substrate pores.

**Figure 6 materials-13-02754-f006:**
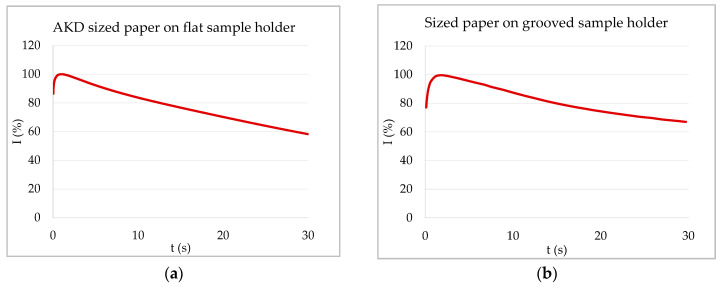
ULP curves of sized paper in contact with water. (**a**) AKD (alkyl ketene dimer) sized paper on flat sample holder—air is entrapped. (**b**) Sized copy paper on grooved sample holder (measurement from [16])—air can leave the sample on the backside via the grooves. Both papers have contact angles of about 100° with water, thus no capillary penetration. The signal decrease is created by fiber swelling, diffusion, or surface transport of liquid into the substrate (in this case usually paper).

**Figure 7 materials-13-02754-f007:**
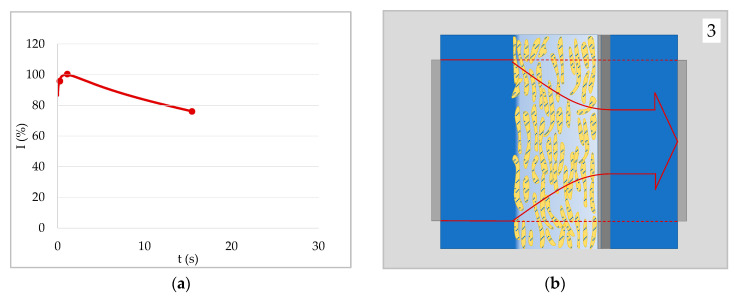
Liquid uptake of a surface hydrophobized paper (no capillary penetration). (**a**) ULP curve of an AKD sized paper in contact with water. (**b**) Graphical illustration of the water uptake into the fibers by vapor and surface diffusion as well as intra-fiber liquid penetration (third stage of liquid uptake; stages one and two are equivalent to Figure 3b, digits 1 and 2). The density increase within the fibers leads to a higher ultrasonic reflection at the fiber-pore interfaces and thus to a decrease of the signal.

**Figure 8 materials-13-02754-f008:**
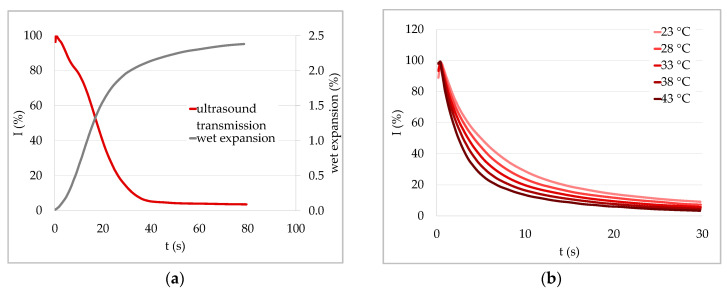
ULP measurement results underlining the effect of fiber swelling on ultrasound transmission. (**a**) Ultrasound transmission intensity I and wet expansion for a coated printing paper in contact with water. Ultrasound transmission and wet expansion are strongly correlated. (**b**) Temperature dependence of ULP measurements of sized copy paper in contact with water. The high temperature dependence indicates a high influence of the vapor pressure and thus diffusive water transport into the fibers (both adapted from [6]).

**Figure 9 materials-13-02754-f009:**
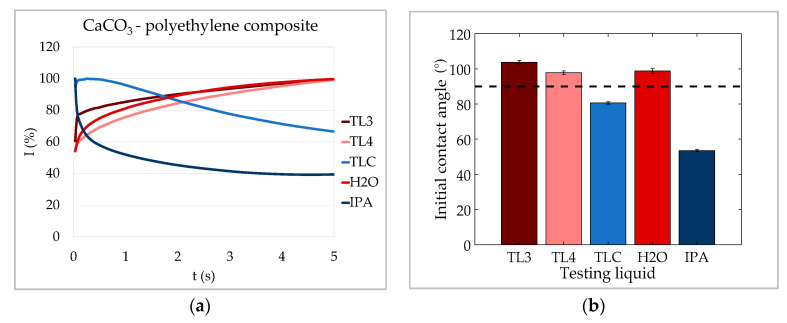
ULP measurements on CaCO_3_ polyethylene composite sheets. (**a**) ULP curves; (**b**) initial contact angles θ (after 40 ms) of the testing liquids on the CaCO_3_ polyethylene composite sheets. Liquids with capillary penetration (θ < 90°) exhibit a decreasing curve (case 2). Liquids without capillary penetration (θ > 90°) show a monotonically increasing curve; the curve is not decreasing because the substrate cannot swell (case 1; error bars indicate 95 percent confidence intervals).

**Figure 10 materials-13-02754-f010:**
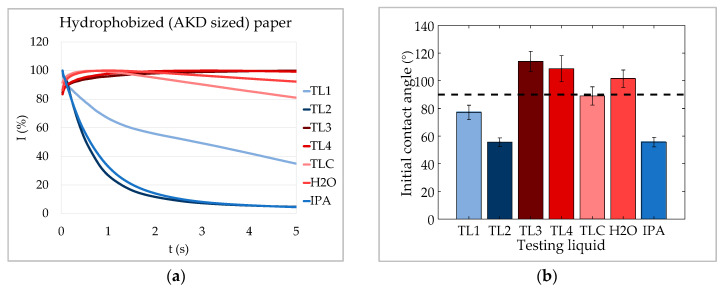
ULP measurements on hydrophobized (AKD sized) paper. (**a**) ULP curves and (**b**) initial contact angles θ (after 20 ms) of the testing liquids on the paper. Liquids with capillary penetration (θ < 90°) exhibit a fast decreasing curve (case 2). Those without capillary penetration (θ > 90°) show a slow decrease after a maximum, which indicates liquid uptake into the fibers due to diffusion and intra-fiber liquid transport (error bars indicate 95 percent confidence intervals.).

**Figure 11 materials-13-02754-f011:**
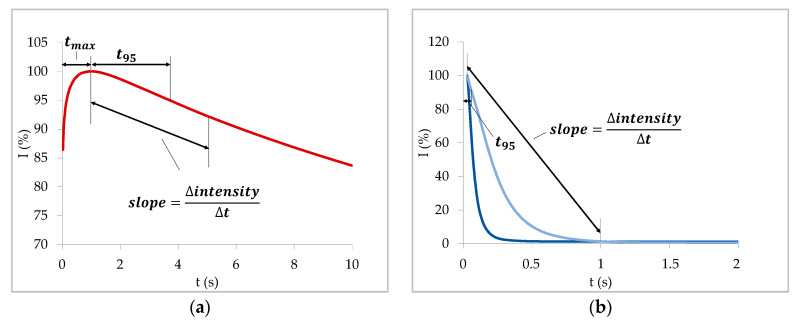
Proposed parameters for the evaluation of ULP measurements. (**a**) Curves with a maximum (case 3): t_max_ = time after which fiber swelling starts; slope/T95 = speed of water uptake of the fibers and (**b**) curves without maximum (case 2): slope/T95 = penetration speed.

**Table 1 materials-13-02754-t001:** Wave impedances for air, water, and pulp fibers.

Medium	Speed of Sound (m/s)	Density (kg/m³)	Wave Impedance (kg/m²s)
air ^1^	343	1.29	442.47
water ^1^	1490	998	1,487,020
fiber ^2^	1493	1500	2,239,548

^1^ values at 20 °C taken from [27]; ^2^ values based on [28].

**Table 2 materials-13-02754-t002:** Reflection coefficients for air, water, and fiber boundaries.

R_air,water_	R_air,fiber_	R_water,fiber_
0.9988	0.9992	0.0408

**Table 3 materials-13-02754-t003:** Test liquids and their components.

Test Liquid	Water (w%)	Glycerol (w%)	Hexanediol (w%)	Isopropanol (w%)	Dye (w%)
TL1	41.9	48	10	-	0.1
TL2	64.9	25	10	-	0.1
TL3	42.4	57.5	-	-	0.1
TL4	74.9	25	-	-	0.1
TLC	56.4	42	1.5	-	0.1
H2O	100	-	-	-	-
IPA	84	-	-	16	-

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
