# Peer review of "Ultrasonic Liquid Penetration Measurement in Thin Sheets—Physical Mechanisms and Interpretation"

_materials, 2020, doi:10.3390/ma13122754_

Round 1
Reviewer 1 Report
This paper report very interesting work on ULP with good description of the mechanisms that can take place
I recommend to include a scheme of the set up described in lines 231-241 like it was done for the cases before. I understand that the second holder has grooves but when it is said.... Thus, the sample is in contact with the tape only at the peaks between the grooves. Still, the sample is in contact with the liquid only on the top side...is totally unclear to me.
Finally on lines 505-506
After some time, the substrate (fibers) starts to take up water via vapor and surface diffusion which causes a density increase and thus higher scattering at the air-fiber interfaces which results in a decreasing signal.
Here it is reported an effect of density but before ( on the line 275 and lines above ) the authors explain the mechanism from reflection...I know the relationship between them but it think should be stated with more consistency in the two section
Author Response
This paper report very interesting work on ULP with good description of the mechanisms that can take place
I recommend to include a scheme of the set up described in lines 231-241 like it was done for the cases before. I understand that the second holder has grooves but when it is said.... Thus, the sample is in contact with the tape only at the peaks between the grooves. Still, the sample is in contact with the liquid only on the top side...is totally unclear to me.
We have added Figure 1c – a drawing of the two sample holders, which should clarify the working principle. In the explanation (now line 265) we are referring to the new figure 1c.
After some time, the substrate (fibers) starts to take up water via vapor and surface diffusion which causes a density increase and thus higher scattering at the air-fiber interfaces which results in a decreasing signal.
Here it is reported an effect of density but before ( on the line 275 and lines above ) the authors explain the mechanism from reflection...I know the relationship between them but it think should be stated with more consistency in the two section
We have changed “scattering” in lines 505-506 to “reflection” to provide consistency of terms.
Reviewer 2 Report
Title: Ultrasonic Liquid Penetration Measurement in Thin Sheets – Physical Mechanisms and Interpretation
The paper describes the application of ultrasonic liquid penetration measurements for the characterization of intrinsic properties of porous sheets. The subject of the paper shall be of interest to readers of ‘Materials.’ Overall the manuscript is written in clear and good English. However, it suffers from some deficiencies in the discussions of the fundamentals of ultrasound transmission. It is recommended to revise the chapter 2. ‘Fundamentals of ultrasound transmission significantly’. Specific comments are the following.
- The chapter does not provide links with existing knowledge and literature on the propagation of ultrasonic waves in layered media.
- The effects of the layer thickness and its variation on the reflection and propagation of ultrasonic waves, which is important for paper layers, are not included in the discussion.
- The phrase ‘Attenuation is mainly caused by viscous losses, reflection occurring at phase boundaries, and scattering,’ perhaps is not correct as thermal losses are not discussed.
- The term ’ Scattering cross-section’ represented in Figure 2 is not defined in the text, and its relation with the terms ‘scattering’ and ‘attenuation’ is not described.
- The terms ‘ultrasound attenuation’ and ‘transmitted signal intensity’ are not related to each other clearly.
- Figure 1 describing the principles of the measurements is confusing. It combines projections in one 2D image. It is recommended to be replaced by ether a 3D image or by two separate 2D cross-sections, clearly illustrating the direction of propagation of the ultrasonic wave and the position of the sample.
Author Response
The paper describes the application of ultrasonic liquid penetration measurements for the characterization of intrinsic properties of porous sheets. The subject of the paper shall be of interest to readers of ‘Materials.’ Overall the manuscript is written in clear and good English. However, it suffers from some deficiencies in the discussions of the fundamentals of ultrasound transmission. It is recommended to revise the chapter 2. ‘Fundamentals of ultrasound transmission significantly’. Specific comments are the following.
The chapter does not provide links with existing knowledge and literature on the propagation of ultrasonic waves in layered media. The effects of the layer thickness and its variation on the reflection and propagation of ultrasonic waves, which is important for paper layers, are not included in the discussion.
The attenuation of the signal is mainly occurring at the interfaces between different materials (discussed in section 2.1 ) and scattering (discussed in section 2.2). Thermal and viscous losses in the material have only a minor impact (see lines 100-105 in the manuscript). Thus the thickness of the material layers and their variation is having only minor effects on the ultrasound transmission. In order to better work this out we have added the following sentence (line 105-106):
“Thus the signal attenuation within a material is not relevant, attenuation is dominated by reflection and scattering at material interfaces.”
The phrase ‘Attenuation is mainly caused by viscous losses, reflection occurring at phase boundaries, and scattering,’ perhaps is not correct as thermal losses are not discussed.
In line 103 we have changed “viscous losses” to “thermal and viscous losses”
The term ’ Scattering cross-section’ represented in Figure 2 is not defined in the text, and its relation with the terms ‘scattering’ and ‘attenuation’ is not described.
We have added a definition for scattering cross section (lines 155-157)
“The scattering cross section thereby refers to the ratio of the total scattered power and the intensity of the incoming signal. The total scattered power is obtained by integrating the scattered intensity in each direction over the surface of a sphere.”
The terms ‘ultrasound attenuation’ and ‘transmitted signal intensity’ are not related to each other clearly.
We have added the following text to the manuscript (lines 100-103)
“On its way through the sample, the ultrasound wave is attenuated, i.e. the amplitude and ultrasound intensity decrease. The ultrasound intensity is defined as the energy flow rate transmitted through a unit area perpendicular to the propagation direction and is proportional to the square of the amplitude [25].”
Figure 1 describing the principles of the measurements is confusing. It combines projections in one 2D image. It is recommended to be replaced by ether a 3D image or by two separate 2D cross-sections, clearly illustrating the direction of propagation of the ultrasonic wave and the position of the sample.
Indeed, the image was confusing. We have changed Figure 1 to a 3D view of the sample.
%.“
Reviewer 3 Report
This manuscript reviewed three possible mechanisms for ultrasonic liquid penetration (ULP) measurements. The manuscript is well presented. There are some comments for the authors to consider. ULP is a specific technique. Some basic requirements are needed to present a clear picture. 1. What are the requirements for the sample to be tested? More specifically, the elastic properties, the porous ratio, and the thickness of the sample should be discussed. 2. What are the liquid used? Are there standard liquids? 3. What is the ultrasound emission signal? Is a pulse or a continuous wave? Why the ultrasound intensity over time is tested? 4. What are the parameters in terms of air and capillary analyses?
Author Response
This manuscript reviewed three possible mechanisms for ultrasonic liquid penetration (ULP) measurements. The manuscript is well presented. There are some comments for the authors to consider. ULP is a specific technique. Some basic requirements are needed to present a clear picture. 1. What are the requirements for the sample to be tested? More specifically, the elastic properties, the porous ratio, and the thickness of the sample should be discussed.
We have added additional text to explain the sample requirements (lines 36-40)
“The only requirement for a sample to be measurable seems to be that attenuation is not too high so that a signal can still be detected at the receiver. This is affected by a combination of factors like sample thickness, porosity, pore size and other material properties like density and elasticity. Furthermore the signal frequency is playing a large role. For samples measured with 2 MHz thickness was between 40µm and 2mm, pore size was between 0.01µm and 100µm and porosity was between 2% and 90%.“
- What are the liquid used? Are there standard liquids?
Water and an isopropanol-water mixture are used most frequently, but there are no real standard liquids. Please refer to lines 32-35 where this is explained.
3.a) What is the ultrasound emission signal? Is a pulse or a continuous wave?
First we have added some text to explain the emission signal (lines 45-55)
“The signal is a pulse with an initial pulse length in the sender of about 1µs. At the receiver, the pulse builds up over seven oscillations and declines over seven oscillations. Consequently, no difference compared to a continuous wave regarding the interference should be seen in practice [24].”
3.b) Why the ultrasound intensity over time is tested?
The change of the ultrasound signal over time is indicating the penetration and diffusion of the testing liquid into the sample over time. Hence the amount of liquid transported to the substrate as a function of time is tested. Please refer to Figures 3 and 4 to illustrate the principle. Captions of
Figures 3 and 4 were revised to better communicate this idea
4. What are the parameters in terms of air and capillary analyses?
We are not entirely sure what the reviewer has meant by this question. We have added some text on the porosity (air content) and pore sizes (capillary size) of the substrates investigated (lines 36-40). For samples measured with 2 MHz measurement the pore size was between 0.01µm and 100µm and porosity was between 2% and 90